# YOU MOSTLY WALK ALONE: ANALYZING FEATURE ATTRIBUTION IN TRAJECTORY PREDICTION

Osama Makansi[†2], Julius von Kügelgen[3,4], Francesco Locatello[1], Peter Gehler[1], Dominik Janzing[1],
Thomas Brox[*1,2], and Bernhard Schölkopf[*1,3]

[1]Amazon, [2]University of Freiburg, [3]Max Planck Institute for Intelligent Systems Tübingen,
[4]University of Cambridge

## ABSTRACT

Predicting the future trajectory of a moving agent can be easy when the past trajectory continues smoothly but is challenging when complex interactions with other agents are involved. Recent deep learning approaches for trajectory prediction show promising performance and partially attribute this to successful reasoning about agent-agent interactions. However, it remains unclear which features such black-box models actually learn to use for making predictions. This paper proposes a procedure that quantifies the contributions of different cues to model performance based on a variant of Shapley values. Applying this procedure to state-of-the-art trajectory prediction methods on standard benchmark datasets shows that they are, in fact, unable to reason about interactions. Instead, the past trajectory of the target is the only feature used for predicting its future. For a task with richer social interaction patterns, on the other hand, the tested models do pick up such interactions to a certain extent, as quantified by our feature attribution method. We discuss the limits of the proposed method and its links to causality.

## 1 INTRODUCTION

Predicting the future trajectory of a moving agent is a significant problem relevant to domains such as autonomous driving (Weisswange et al., 2021; Xu et al., 2014), robot navigation (Chen et al., 2018), or surveillance systems (Morris & Trivedi, 2008). Accurate trajectory prediction requires successful integration of multiple sources of information with varying signal-to-noise ratios: whereas the target agent's past trajectory is quite informative most of time, interactions with other agents are typically sparse and short in duration, but can be crucial when they occur.

Graph neural networks (N. Kipf & Welling, 2017) and transformers (Vaswani et al., 2017) have led to recent progress w.r.t. average prediction error (Mohamed et al., 2020; Yu et al., 2020; Salzmann et al., 2020; Mangalam et al., 2020), but the predicted distributions over future trajectories remain far from perfect, particularly in unusual cases. Moreover, the black-box nature of state-of-the-art models (typically consisting of various recurrent, convolutional, graph, and attention layers) makes it increasingly difficult to understand what information a model actually learns to use to make its predictions.

To identify shortcomings of existing approaches and directions for further progress, in the present work, we propose a tool for *explainable trajectory prediction*. In more detail, we use Shapley values (Shapley, 1953; Lundberg & Lee, 2017) to develop a feature attribution method tailored specifically to models for multi-modal trajectory prediction. This allows us to quantify the contribution of each input variable to the model's *performance* (as measured, e.g., by the negative log likelihood) both locally (for a particular agent and time point) and globally (across an entire trajectory, scene, or dataset). Moreover, we propose an aggregation scheme to summarize the contributions of all neighboring agents into a single *social interaction score* that captures how well a model is able to use information from interactions between agents.

---

[†]Work done during an internship at Amazon. Contact email: makansio@cs.uni-freiburg.de
[*]Equal contribution.

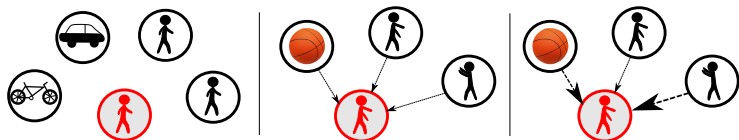

Figure 1: Our proposed method allows for comparing different trajectory prediction models in terms of the extent to which they use social interactions for making predictions (left: none, middle: weak, right: strong). The target agent, whose future trajectory is to be predicted, is shown in red, and modelled interactions are represented by arrows with width indicating interaction strength.

By applying our analysis to representative state-of-the-art models, Trajectron++ (Salzmann et al., 2020) and PECNet (Mangalam et al., 2020), we find that—contrary to claims made in the respective works—the predictions of these models on the common benchmark datasets ETH-UCY (Pellegrini et al., 2009; Leal-Taixé et al., 2014), SDD (Robicquet et al., 2016), and nuScenes (Caesar et al., 2020) are actually *not* based on interaction information. We, therefore, also analyze these models' behavior on an additional sports dataset SportVU (Yue et al., 2014) for which we expect a larger role of interaction. There, models do learn to use social interaction, and the proposed feature attribution method can quantify (i) when these interaction signals are active and (ii) how relevant they are, see Fig. 1 for an overview.

Overall, our analysis shows that established trajectory prediction datasets are suboptimal for benchmarking the learning of interactions among agents, but existing approaches do have the capability to learn such interactions on more appropriate datasets. We highlight the following contributions:

- We address, for the first time, feature attribution for trajectory prediction to gain insights about the actual cues contemporary methods use to make predictions. We achieve this by designing a variant of Shapley values that is applicable to a large set of trajectory prediction models (§ 3).
- With this method, we quantify feature attributions both locally (per scenario; § 3.1) and globally (over the whole dataset; § 3.2) and study the robustness of given models (§ 3.3).
- Our study uncovers that, on popular benchmarks, existing models do not use interaction features. However, when those features have a strong causal influence on the target's future trajectory, these models start to reason about them (§ 4.4).

## 2    BACKGROUND & RELATED WORK

Our primary goal is to better understand the main challenges faced by existing trajectory prediction models. First, we formalize the problem setting (§ 2.1) and review prior approaches (§ 2.2), as well as explainability tools (§ 2.3), which we will use to accomplish this goal in § 3.

### 2.1    THE TRAJECTORY PREDICTION PROBLEM SETTING

Let $\{X_t^1, ..., X_t^n\}, t \in \mathbb{Z}$ denote a set of time series corresponding to the trajectories of $n$ agents that potentially interact with each other; see Fig. 2 (a) for an illustration. All $X_t^i$ are assumed to take values in $\mathbb{R}^d$, that is, the state of agent $i$ at time $t$ is a vector $x_t^i \in \mathbb{R}^d$ encoding, e.g., its 2D position, velocity, and acceleration. We refer to an observed temporal evolution $x_{1:T}^i = (x_1^i, ..., x_T^i)$ of a single agent as a *trajectory* of length $T$ and to the collection of trajectories $x_{1:T}^{1:n} = (x_{1:T}^1, ..., x_{1:T}^n)$ of all $n$ agents as a *scene*. Assume that we have access to a training dataset consisting of $M$ such scenes.[1]

The trajectory prediction task then consists of predicting the future trajectory $(X_{t+1}^i, ..., X_{t+\Delta t}^i)$ of a given target agent $i$ and at a given time $t$ up to a time horizon $\Delta t$ given the observed past $x_{t-h:t}^{1:n}$ of both the target agent itself and of all the other agents, where $h$ is the length of the history that is taken into account. Formally, we thus aim at learning the distributions $\mathbb{P}(X_{t+1:t+\Delta t}^i | X_{t-h:t}^{1:n})$ for any $t \in \mathbb{Z}$ and $i \in N := \{1, ..., n\}$ with given $\Delta t$ and $h$.

Since any given agent (e.g., a particular person) typically only appears in a single scene in the training dataset, solving the trajectory prediction task requires generalizing across scenes and agent

---

[1]In practice, scenes may differ both in length and in the number of present agents; for the latter, we define $n$ as the maximum number of agents present in a given scene across the dataset and add "dummy" agents to scenes with fewer agents, see § 4.1 for further details.

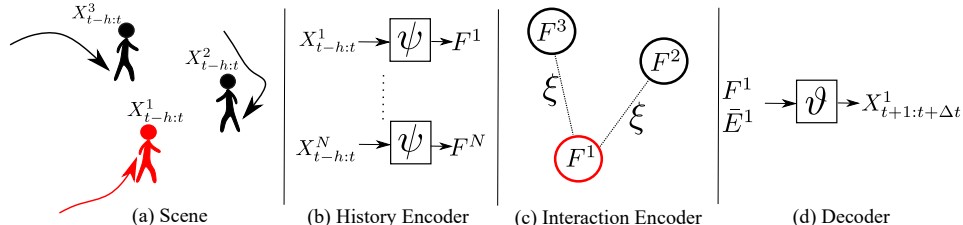

(a) Scene   (b) History Encoder   (c) Interaction Encoder   (d) Decoder

Figure 2: General framework of existing trajectory prediction methods: Given (a) a scene with dynamic agents, (b) the history of every agent is encoded as $F^i$, and (c) edges to other agents are learned by an interaction encoder and aggregated to form the overall edge features $\bar{E}^i$. Finally, (d) both the history and interaction features are passed to a decoder to predict the future trajectory.

identities. In other words, it is necessary to learn how an agent's future trajectory, *in general*, depends on its own past and on the past behavior of other neighboring agents.

## 2.2 EXISTING APPROACHES FOR TRAJECTORY PREDICTION

Fig. 2 (b)–(d) shows a general framework that unifies most existing approaches for trajectory prediction. It consists of three main modules:

(b) a *history encoder* $\psi$ that learns an embedding $F^i = \psi(X^i_{t-h:t})$ of the history of each agent;

(c) an *interaction encoder* $\xi$ that incorporates information from neighboring agents by learning an embedding for the edge between two agents $E^{ij} = \xi(X^i_{t-h:t}, X^j_{t-h:t})$; and

(d) a *decoder* $\vartheta$ that combines both history and interaction features to generate the predicted future trajectory $\hat{x}^i_{t+1:t+\Delta t} := \vartheta(F^i, \bar{E}^i)$ of the target agent, where $\bar{E}^i$ is the result of aggregating all edge embeddings (see below).

Existing approaches mostly differ in the choices of these modules. To handle the temporal evolution of the agent state, LSTMs (Hochreiter & Schmidhuber, 1997) are widely used for $\psi$ and $\vartheta$, i.e., to encode the history and decode the future trajectory, respectively (Alahi et al., 2016; Zhang et al., 2019). Since the future is highly uncertain, stochastic decoders are typically used to sample multiple trajectories, $\hat{x}^i_{t+1:t+\Delta t} \sim \vartheta(.)$, e.g., using GANs (Gupta et al., 2018) or conditional VAEs (Lee et al., 2017; Salzmann et al., 2020). Alternatively, Makansi et al. (2019) proposed a multi-head network that directly predicts the parameters of a mixture model over the future trajectories.

To handle interactions between moving agents (Fig. 2c), existing works model a scene as a graph in which nodes correspond to the state embeddings $F^i$ of the $n$ agents and edges are specified by an adjacency matrix $A$ with $A^{ij} = 1$ iff. agents $i$ and $j$ are considered neighbors (based, e.g., on their relative distance). Given this formulation, local social pooling layers have been used to encode and aggregate the relevant information from nearby agents within a specific radius (Alahi et al., 2016; Gupta et al., 2018). More recently, Mangalam et al. (2020) proposed a non-local social pooling layer, which uses attention and is more robust to false neighbor identification, while Salzmann et al. (2020) modeled the scene as a directed graph to represent a more general set of scenes and interaction types, e.g., asymmetric influence. In a different line of work, Mohamed et al. (2020) proposed undirected spatio-temporal graph convolutional neural networks to encode social interactions, and Yu et al. (2020) additionally incorporated transformers based on self-attentions to learn better embeddings. In general, the edge encodings $\{E^{ij}\}_j$ for agent $i$ are then aggregated over the set of its neighbors (i.e., those $j$ with $A^{ij} = 1$) to yield the interaction features $\bar{E}^i$, e.g., by averaging as follows:

$$\bar{E}^i = \frac{\sum_{j \neq i} A^{ij} E^{ij}}{\sum_{j \neq i} A^{ij}} . \tag{1}$$

Although some of the above works show promising results on common benchmarks, it is unclear what information these methods actually use to predict the future. In the present work, we focus mainly on this aspect and propose an evaluation procedure that quantifies the contribution of different features to the predicted trajectory, both locally and globally.

## 2.3 Explainability and Feature Attribution

An important step toward interpretability in deep learning are *feature attribution* methods which aim at quantifying the extent to which a given input feature is responsible for the behavior (e.g., prediction, uncertainty, or performance) of a model. Among the leading approaches in this field is a concept from cooperative game theory termed *Shapley values* (Shapley, 1953), which fairly distributes the payout of a game among a set of $n$ players.[2] Within machine learning, Shapley values can be used for feature attribution by mapping an input $x = (x_1, ..., x_n)$ to a game in which players are the individual features $x_i$ and the payout is the model behavior $f : \mathcal{X}_1 \times ... \times \mathcal{X}_n \to \mathbb{R}$ on that example (Lundberg & Lee, 2017). Formally, one defines a set function $\nu : 2^N \to \mathbb{R}$ with $N := \{1, ..., n\}$ whose output $\nu(S)$ for a subset $S \subset N$ corresponds to running the model on a modified version of the input $x$ for which features not in $S$ are "dropped" or replaced (see below for details). The contribution of $x_i$, as quantified by its Shapley value $\phi(x_i)$, is then given by the difference $\nu(S \cup \{i\}) - \nu(S)$ between including and not including $i$, averaged over all subsets $S$:

$$\phi(x_i) = \sum_{S \subseteq N \setminus \{i\}} \frac{1}{n \binom{n-1}{|S|}} \left( \nu(S \cup \{i\}) - \nu(S) \right). \tag{2}$$

There are different variants of Shapley values based mainly on the following two design choices:

 (i) *What model behavior (prediction, uncertainty, performance) is to be attributed?* (choice of $f$)
(ii) *What is the reference baseline, that is, how are features dropped?* (choice of $\nu$)

A common choice for (i) is the output (i.e., prediction) of the model. Alternatively, Janzing et al. (2020a) defined $f$ as the uncertainty associated with the model prediction, thus quantifying to what extent each feature value contributes to or reduces uncertainty. In the present work, we will focus on attributing model performance. As for (ii) the set function $\nu$, Lundberg & Lee (2017) originally proposed to replace the dropped features $x_{N \setminus S}$ with samples from the conditional data distribution given the non-dropped features: $\nu(S) = \mathbb{E}[f(x_S, X_{N \setminus S})|X_S = x_S]$; they then approximate this with the marginal distribution, $\nu(S) = \mathbb{E}[f(x_S, X_{N \setminus S})]$, using the simplifying assumption of feature independence. Janzing et al. (2020b) argued from a causal perspective that the marginal distribution is, in fact, the right distribution to sample from since the process of dropping features naturally corresponds to an *interventional* distribution, rather than an observational (conditional) one. In an alternative termed *baseline* Shapley values, the dropped features are replaced with those of a predefined baseline $x'$: $\nu(S) = f(x_S, x'_{N \setminus S})$, see Sundararajan & Najmi (2020) for details.

## 3 Explainable Trajectory Prediction

In the present work, we are interested in better understanding what information is used by trajectory prediction models to perform well, that is, quantifying the contribution of each input feature to the *performance* of a given model (rather than to its actual output). Therefore, we define the behavior $f$ to be attributed (i.e., choice (i) in § 2.3) as the error of the output prediction $f := L(\hat{x}^i_{t+1:t+\Delta t}, x^i_{t+1:t+\Delta t})$, where $L$ denotes any loss function (see § 4.3 for common choices).

### 3.1 Which Shapley Value Variant: How to Drop Features?

Next, we need to decide how to define the set function $\nu$ (choice (ii) in § 2.3), i.e., how to drop features. Since this choice can be highly context-dependent, we need to ask: what is the most reasonable treatment for dropped features *in the context of trajectory prediction*? Intuitively, dropping a subset of agents should have the same effect as if they were not part of the scene to begin with: ideally, we would like to consider the behavior difference between agents being present or not.

As discussed in § 2.3, Shapley values are commonly computed by replacing the dropped features with random samples from the dataset or with a specific baseline value $x'$. For the latter, choosing the baseline value $x'$ is often non-trivial: for instance, Dabkowski & Gal (2017) use the feature-wise mean as a baseline, while Ancona et al. (2019) set the baseline to zero. However, for either choice the dropped features still contribute some signal (e.g., causing gray-ish or black pixels, respectively,

---

[2]Shapley values are the only method that satisfy a certain set of desirable properties or axioms, see (Lundberg & Lee, 2017) for details. For other feature attribution methods, we refer to Appendix A.

in the context of images) that may affect the feature attribution scores. The same caveat applies to using random samples which—in addition to the computational overhead—also makes this variant unattractive for our needs.

For trajectory prediction, we thus opt for a modified version of baseline Shapley values where we choose the baseline to be a *static, non-interacting* agent. This means that the contribution of the past trajectory of a target agent is quantified relative to a static non-moving trajectory, and the contributions of neighboring agents are quantified relative to the same scene after removing those agents. Owing to the flexible formulation of the interaction encoder as a graph (see § 2.2 and Fig. 2c), dropping a neighboring agent from the scene is equivalent to cutting the edge between the target agent and that neighbor, and thus easy to implement. Formally, we achieve this by modifying the adjacency matrix of the graph used in (1) as follows:

$$A^{ij} = \begin{cases} 0, & \text{if } j \notin S \\ A^{ij}, & \text{otherwise} \end{cases}. \tag{3}$$

This formulation applies to most existing frameworks using directed (Salzmann et al., 2020) or undirected graphs (Alahi et al., 2016; Gupta et al., 2018; Mangalam et al., 2020; Mohamed et al., 2020).

## 3.2 Contribution Aggregation: the Social Interaction Score

With the above formulation, quantifying the contribution of features (i.e, the past trajectory of the agent-of-interest and the interaction with other agents) in a specific scenario is possible. On the other hand, aggregating these Shapley values over multiple scenarios is important to draw conclusions about the behavior of the model on a specific dataset. Formally, given the Shapley values of a feature for a specific scenario $\phi(x_i)$, we aim at quantifying the contribution of the feature over the whole dataset $\phi_i$. The common methodology is to average the obtained Shapley values for every feature over multiple scenarios. This is problematic when features vary over scenarios. For instance, the neighbors of the agent-of-interest may change over time and are different over scenes, thus aggregating their contribution over the whole dataset yields the wrong statistics.

To address this problem, we split the features into two types: permanent features $\mathcal{P}$ and temporary features $\mathcal{T}$. The past trajectory of the agent-of-interest is an example of the former type, while the neighboring agents are examples of the latter. For every permanent feature, we use the common average aggregation to estimate the overall contribution of that feature $\phi_i = \mathbb{E}[\phi(x_i)]$. For the temporary features, we first aggregate them locally (within the same scenario) and then globally. For the global aggregation, we use the average operator. Since interactions with neighbors are usually sparse (i.e., only few neighbors have true causal influence on the target), we choose the max operator for the local aggregation. In other words, we quantify the contribution of the most influential neighboring agent, which we call the *social interaction score*:

$$\phi_{\mathcal{T}} = \mathbb{E}[\max_{j \in \mathcal{T}}(\phi(x_j))]. \tag{4}$$

A larger non-negative score indicates that at least one of the neighbors has a significant positive contribution to the predicted future while a zero score indicates that none of the neighbors are used.

## 3.3 Robustness Analysis

Beside the contribution analysis of existing features (such as neighboring agents) via our formulation of the Shapley values, we propose to use the same methodology to study the robustness of pretrained models when random agents are added to the scene. Thanks to the structure of the interaction encoder (modeled as a graph), adding a random neighbor is equivalent to adding a new incoming edge in Fig. 2 (c). The random agent can be drawn from the same scene at a different time or from another scene. Random agents should not contribute to the future of the agent-of-interest,[3] thus their Shapley values given a robust model should be close to zero; see the dummy property of Shapley values (Sundararajan & Najmi, 2020). In other words, this analysis tests the model ability in distinguishing between real and random neighbors.

---

[3]In the context of pedestrian navigation, a random agent may still play a role for avoiding collisions. Using some rules for the random agent (e.g., not in close proximity to the target agent) may alleviate such concerns.

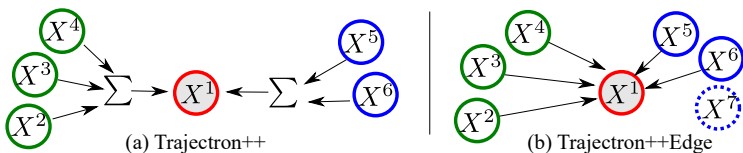

Figure 3: The interaction encoder of Trajectron++ (a) and our Trajectron++Edge (b). Instead of learning a single edge to all neighbors of the same type (i.e., to all pedestrians), we allow the model to learn separate edges to existing neighbors and add dummy agents to handle the variable number of neighbors (dashed circle). Here, red denotes the target agent and neighbors are colored according to their type (e.g., green for pedestrians, blue for vehicles).

## 4 EXPERIMENTS

Given the proposed method, we analyze the role of interaction both locally and globally for different state-of-the-art models in trajectory prediction.

### 4.1 MODELS

Our analysis is based on three recent state-of-the-art models: Social-STGCNN (Mohamed et al., 2020), Trajectron++ (Salzmann et al., 2020) and PECNet (Mangalam et al., 2020). We refer to § 2.2 for details about the implementation choices of the key modules (history/interaction encoders and decoder). For PECNet, we compare the three provided pretrained models (Mangalam et al., 2020).

Alternatively, we propose a new variant of the Trajectron++ (named Trajectron++Edge) in which we make both the history and interaction encoders stronger. For the history encoder (LSTM), we make it one layer deeper. To handle a variable number of neighboring agents, they propose to aggregate all neighbors from the same type (e.g., all pedestrians) into one node and learn a single edge to the target agent; see Fig. 3 (a). Alternatively, we propose to learn separate edges to all neighboring agents and to add "dummy" agents to the scene until we reach a predefined maximum number; see Fig. 3 (b). Notably, giving the network the freedom to learn separate edges between any two agents is more powerful. For instance, two neighboring pedestrians could have different effects on the future of the target agent. Using our evaluation procedure, we can compare the two variants by quantifying the contribution of both features (history and interaction).

### 4.2 DATASETS

**ETH-UCY** is one of the most common benchmarks for trajectory prediction. The dataset is a combination of the ETH (Pellegrini et al., 2009) and UCY (Leal-Taixé et al., 2014) datasets with 5 different scenes and $1,536$ pedestrians. We use the standard 5-fold cross validation for our analysis.

**Stanford Drone Dataset (SDD)** (Robicquet et al., 2016) is a large dataset with 20 different scenes covering multiple areas at Stanford university. It consists of over $11k$ agents of different types (e.g, pedestrians, bikers, skaters, cars, etc.). We follow the standard train/test split used in previous works (Mangalam et al., 2020; Gupta et al., 2018).

**nuScenes** (Caesar et al., 2020) is one of the largest autonomous driving datasets with more than $1,000$ scenes collected from Boston and Singapore, where each is 20 seconds long. The dataset also has semantic maps with 11 different layers such as drivable areas and pedestrian crossings. We also use the standard training/testing splits (Salzmann et al., 2020).

**SportVU** (Yue et al., 2014) is a tracking dataset for games recorded from multiple seasons of the NBA. Every scene has two teams of 5 players and the ball. We pre-process the dataset to remove short scenes and randomly select 7,000 scenes for training and 100 scenes for testing.

### 4.3 EVALUATION METRICS

**min-ADE** is the minimum over all predicted trajectories of the *average* displacement error (L2 distance) to the ground truth. Similarly, **min-FDE** is the minimum *final* displacement error, i.e., the minimum L2 distance over all predicted trajectories to the ground truth end point at time $t + \Delta t$.

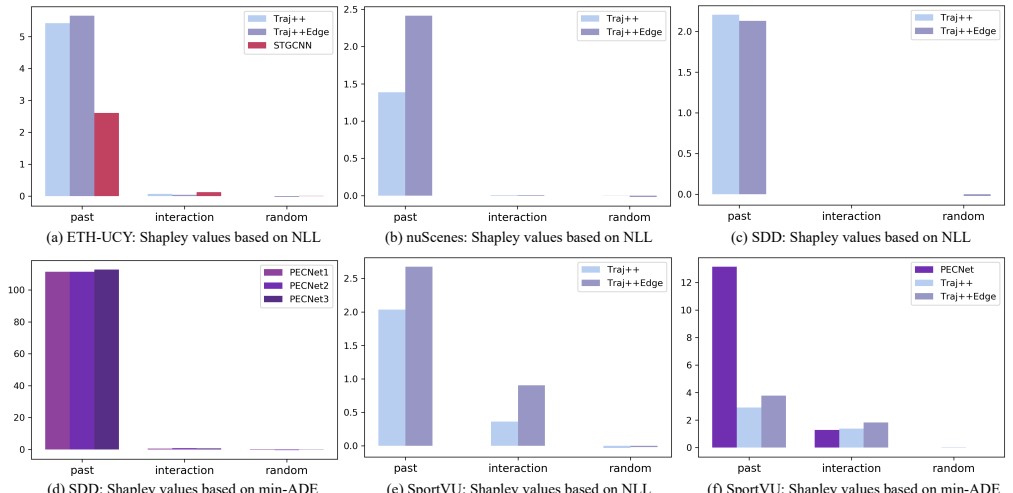

Figure 4: Shapley values of Trajectron++ variants and PECNet on the three common benchmarks for future trajectory (a-d) and the SportVU dataset (e,f). Notably, the contribution of social interactions (neighbors) is almost zero on the common benchmarks, while neighbors do play an important role when predicting the future trajectory on the SportVU dataset.

The previous two metrics are biased since they report only the minimum error over many predictions and are commonly used for testing multi-modality of the prediction. To assess all predicted trajectories for those methods predicting a multimodal probabilistic distribution of the future, we report also the **NLL** of the ground truth given the predicted distribution.

For the Shapley values, we choose the NLL of the model as the output of the set function $\nu$ for both variants of Trajectron++ and Social-STGCNN. On the other hand, PECNet is a non-probabilistic approach and can only output multiple trajectories by decoding multiple samples drawn from the latent space. Therefore, we choose the min-ADE of the model when evaluating PECNet models.

## 4.4 RESULTS

**Analysis on common benchmarks.** As discussed in § 3, we report the contribution for the past trajectory (since it is a permanent feature) and the neighboring agent with the maximum contribution (our social interaction score). Additionally, we also compare to the contribution of a random agent added to the scene (robustness). From the results shown in Fig. 4 on ETH-UCY (a), nuScenes (b) and SDD (c-d), we observe the following: (1) the past trajectory has the largest contribution on the predicted future, (2) the contribution of the neighbors (our *social interaction score*) is insignificant and close to zero, and (3) the contribution of a random neighbor is almost identical to existing neighbors. Notably, for those datasets with heterogeneous agents (e.g, pedestrians, vehicles, bikers, etc.), we report the average results over different types of agents. From the last two observations, we conclude that recent methods are unable to exploit information from neighboring agents to predict the future on these common benchmarks.

**Analysis on the SportVU dataset.** We conduct the same analysis on a dataset with rich patterns of interactions and show the results in Fig. 4 (e, f). Although the past trajectory still has the largest contribution, neighboring agents start to play an important role by contributing to the predicted future. Moreover, there is a significant difference between the maximum contribution of neighbors (our *social interaction score*) and the contribution of a random neighbor.

Our analysis is not only important to explain and test models but also to compare them. As can be seen in Fig. 4 (e), the contributions of both the past trajectory and the neighbors of our Trajectron++Edge variant are significantly larger than the original model. Referring to § 4.1, both the history and interaction encoders are stronger in our variant which explains the increase in their contributions. On the other hand, both variants are robust to random neighbors added to the game. In Fig. 4 (f), we compare PECNet and both variants of Trajectron++ on the SportVU dataset where

Table 1: Average errors (min-ADE / min-FDE / NLL) on trajectory prediction benchmarks. For every model, we show the errors with and without (w/o) interaction, as well as their relative difference.

| Method \ Dataset | ETH-UCY | SDD | nuScenes | SportVU |
|---|---|---|---|---|
| Social-STGCNN | 0.45 / 0.76 / -1.05 | - | - | - |
| w/o interaction | 0.45 / 0.76 / -0.95 | - | - | - |
| Diff | 0.0 / 0.0 / -0.1 | - | - | - |
| PECNet | - | 9.29 / 15.93 / - | - | 7.95 / 17.38/ - |
| w/o interaction | - | 9.28 / 15.93 / - | - | 9.78 / 17.38/ - |
| Diff | - | 0.0 / 0.0 / - | - | -1.83 / 0.0 / - |
| Traj++ | 0.30 / 0.51 / -0.33 | 1.35 / 2.05 / 1.76 | 0.49 / 0.77 / 0.52 | 4.86 / 5.31 / 5.99 |
| w/o interaction | 0.31 / 0.52 / -0.33 | 1.35 / 2.07 / 1.75 | 0.49 / 0.77 / 0.50 | 6.62 / 8.98 / 6.77 |
| Diff | -0.01 / -0.01 / 0.0 | 0.0 / -0.02 / 0.01 | 0.0 / 0.0 / 0.02 | -1.76 / -3.67 / -0.78 |
| Traj++Edge | 0.30 / 0.49 / -0.39 | 1.39 / 2.14 / 1.98 | 0.26 / 0.43 / -1.28 | 4.78 / 5.22 / 5.97 |
| w/o interaction | 0.30 / 0.50 / -0.41 | 1.39 / 2.16 / 1.98 | 0.26 / 0.44 / -1.26 | 7.49 / 11.18 / 7.19 |
| Diff | 0.0 / -0.01 / -0.02 | 0.0 / -0.02 / 0.0 | 0.0 / -0.01 / -0.02 | - 2.71 / -5.96 / -1.22 |

we observe that our Trajectron++Edge has the largest social interaction score while PECNet is largely based on the past.

Similarly to our social interaction score, Tab. 1 shows the relative difference in the average error when dropping all neighbors.[4] Here we also observe that social interaction is only important on the SportVU dataset. Instead of dropping all neighbors during inference, we drop them during training and report the difference in Appendix C.

**Local analysis.** In Fig. 5, we show the results of our Trajectron++Edge on two interesting scenarios. Here, every player (and the ball) is represented as a circle and colored according to the team. We plot the past trajectories of all players (solid lines), the ball (yellow) and the future trajectory of the target player (dashed line). We overlay the predicted distribution of the model (blue heatmap) on top of the scene. Additionally, we show the Shapley values of the scenario (right) for the past (red), neighboring players and the ball. For the first scenario (a), we clearly see that the past is the main feature contributing to the predicted future where all other players have insignificant contributions. On the other hand, for the second scenario where interaction with other players is critical to make the correct prediction (b), we can identify that the three main players (counter attacking) and the ball contribute positively to the predicted future, while the past has relatively small influence on the prediction. Interestingly, agent 6 moves to the right (opposite of the true future of the target) and thus has a negative contribution on the prediction of the target agent. Overall, our method is able to identify and visualize which feature has a large influence on the target agent future for specific local events.

## 5 DISCUSSION

**Links to Granger causality.** Although the *relevance* of features *for predicting* the target should not be confused with their *causal impact*, the two are tightly linked. According to the seminal work of Granger (1969), a time series $X_t$ causally influences a target $Y_t$ whenever the past of $X$ helps better predict $Y_t$ from its own past and that of all features other than $X$. This reasoning is justified—and follows from the causal Markov condition and causal faithfulness, see (Peters et al., 2017, Thm. 10.3)—whenever the following three assumptions are satisfied: (i) every feature influences others only with a nonzero time lag (i.e., no instantaneous influence); (ii) there are no latent (unobserved) common causes of the observed features (causal sufficiency); and (iii) the prediction is optimal in the sense that it employs the full statistical information from the past of the features under consideration. Although these assumptions are hard to satisfy *exactly* in practice, Granger causality has been widely applied since decades due to its simplicity. Since our attribution analysis is based on comparing model *performance* (as opposed to *predictions*) with and without a certain feature, it is loosely related to Granger causality. The main differences are that we (a) perform a post-hoc analysis on a single model, rather than training separate ones with and without the feature of interest; and (b) average over subsets of the remaining features, instead of always including them.

---

[4]Our numbers for PECNet on SDD are obtained by retraining their model using their source code. On Traj++, our numbers are obtained after fixing an issue with the derivative computation and is reported to the corresponding authors. Moreover, the reported results on SDD for PECNet and Traj++ are not directly comparable as the errors are in different domains (in pixels for PECNet and in meters for Traj++).

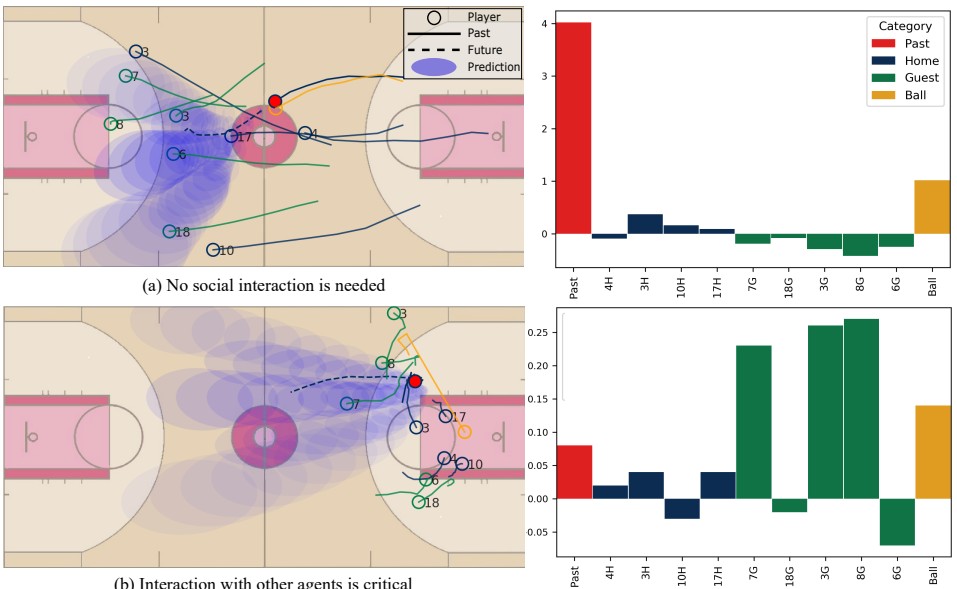

Figure 5: Qualitative analysis of our variant Trajectron++Edge for a scenario where the role of interaction is minimal (a) and a scenario where the role is important (b). For both scenarios, we show the scene with players, the predicted distribution of the future after a fixed $\Delta t$ in blue (left), and the Shapley values of all players, the ball and the target's past trajectory (right). In red is the target agent, the ball is in yellow, and the two teams' other players are in blue/green. Solid lines are the observed past trajectory and the dashed line is the ground truth future.

**Interpretation and limitations.** While assumptions (i) and (ii) above describe the general scenarios for which our attribution method may allow a causal interpretation, assumption (iii) points to an important limitation. Since a model simply may not have learned to pick up and make use of certain influences, *we cannot conclude from small Shapley values that no causal influence exists*. If, however, we know a priori that a causal influence does exist and the corresponding Shapley value is (close to) zero, then this points to a failure of the model. Similarly, if there is a causal influence and the model can pick this up, we will see it in the Shapley value. For the trajectory prediction task, where only past information is used for prediction, positive Shapley values thus suggest the existence of a causal influence, subject to (i) and (ii).

**Exclusion vs randomization of features.** As opposed to *detecting* causal influence, we note that *quantifying* its strength via the degree of predictive improvement is not always justified, contrary to what is often believed. Janzing et al. (2013, Sec. 3.3 and Example 7) argued, based on a modification of a paradox described by Ay & Polani (2008), that assessing causal strength by *excluding* the feature for prediction is flawed and showed that it should be *randomized* instead. However, our experiments in Appendix B show only a minor difference between exclusion and randomization in our case.

## 6 CONCLUSION

We addressed feature attribution for trajectory prediction by analyzing to what extent the available cues are actually used by existing methods to improve predictive performance. To this end, we proposed a variant of Shapley values for quantifying feature attribution, both locally and globally, and for studying the robustness of given models. Subject to the assumptions and caveats discussed in § 5, our attribution method can be interpreted as a computationally efficient way of (approximately) quantifying causal influence in the context of trajectory prediction, both for indicating the strength of causal influence of a dataset and for benchmarking how well a particular model uses such influence for its prediction. Using our method, we reveal that existing methods, contrary to their claims, do not rely on interaction features when trained on popular data sets, but that such information is used on other data sets such as SportVU where interactions play a larger role, see Appendix D for further discussion of our finding.

# 7 ACKNOWLEDGMENTS

We would like to thank Lenon Minorics and Atalanti Mastakouri for the useful discussions and feedback. This work was supported by the German Federal Ministry of Education and Research (BMBF): Tübingen AI Center, FKZ: 01IS18039B, and by the Machine Learning Cluster of Excellence, EXC number 2064/1 – Project number 390727645. We also thank Amazon Web Services for providing enough resources to conduct the experiments.

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

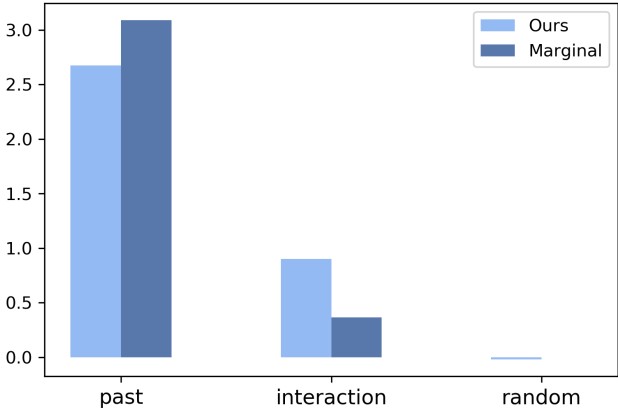

Figure 6: Comparison between the baseline Shapley values (used in our paper) and the marginal Shapley values for our Trajectron++Edge on the SportVU dataset.

## A  OTHER FEATURE ATTRIBUTION METHODS

Other feature attribution methods can generally be categorized into two streams: perturbation-based and gradient-based approaches. The former perturb some parts of the input and use the change in the output as a measure of feature relevance (Zeiler & Fergus, 2014). Such methods are sensitive to the applied perturbation and violate the *sensitivity* axiom (Sundararajan et al., 2017). Gradient-based approaches estimate the feature attributions by investigating the local gradients of the model. To address the saturation problem of gradients, DeepLIFT (Shrikumar et al., 2017) and LRP (Binder et al., 2016) approximate gradients with discrete differences, which results in violating the *implementation invariance* axiom (Sundararajan et al., 2017). Sundararajan et al. (2017) proposed the Integrated Gradients (IG) to estimate the feature attribution relevant to a baseline by integrating over the gradient along the path between the given input and the baseline, which satisfy all required axioms and can be seen as a variant of the Shapley values (Sundararajan & Najmi, 2020).

## B  EXCLUSION VS RANDOMIZATION OF FEATURES

In Fig. 6, we compare the obtained Shapley values using our method and the ones obtained by running the marginal Shapley values. For the latter, we replace the feature by a random feature drawn from the marginal distribution. In other words, instead of removing a neighboring agent, we replace it by a random agent from another game and average over multiple replacements. Clearly, we draw the same conclusions (as in § 4.4) using the marginal Shapley values while our method is more computationally efficient since we do not need to marginalize over many samples.

## C  RETRAINING WITHOUT INTERACTION

To further support our findings, we conduct an additional experiment where we train one of the SOTA models from scratch on a dataset without interaction by removing neighbors to assess if models need interaction or not. We selected the Traj++ model, retrained it on the ETH-UCY dataset, and compared the results to the same model trained with interactions. Tab. 2 shows that both models (with and without interaction) yield the same error, which is consistent with our findings.

## D  FURTHER DISCUSSION

Here we provide two complementary explanations for our finding that existing models behave differently between the popular benchmarks (e.g., ETH-UCY) and the SportVU dataset: (i) a large difference in the *frequency* of interactions, and (ii) a large difference in the *nature* of interactions. As

Table 2: Average errors (min-ADE / min-FDE) for the Traj++ model on ETH-UCY. We observe almost no difference between training with/without interaction with the ETH scene to be an exception where training without interaction yields lower errors.

| Method \Scene | ETH | HOTEL | UNIV | ZARA1 | ZARA2 | AVG |
|---|---|---|---|---|---|---|
| Traj++ | 0.61 / 1.02 | 0.19 / 0.28 | 0.30 / 0.54 | 0.24 / 0.42 | 0.18 / 0.31 | **0.30 / 0.51** |
| Retrained w/o interaction | 0.58 / 0.96 | 0.19 / 0.27 | 0.30 / 0.54 | 0.25 / 0.42 | 0.17 / 0.31 | **0.30 / 0.50** |
| Diff | 0.03 / 0.06 | 0.0 / 0.01 | 0.0 / 0.0 | -0.01/ 0.0 | 0.01 / 0.0 | 0.0 / 0.01 |

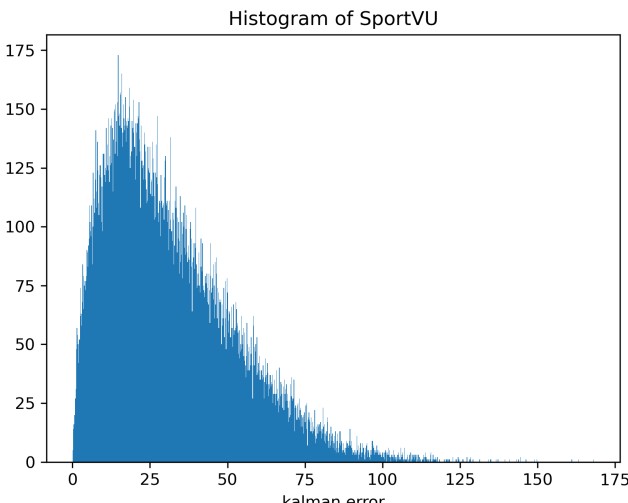

Figure 7: Histogram of the errors of the linear Kalman filter on the SportVU dataset. The fraction of challenging scenarios (i.e., with large kalman errors) are much larger than on the ETH-UCY, see Figure 1 of (Makansi et al., 2021).

for (i), the concurrent work (Makansi et al., 2021)highlights the long-tailed distribution of common pedestrian benchmarks (e.g., ETH-UCY) meaning that most scenarios can be predicted well using a simple linear Kalman filter (so-called easy scenarios) while more challenging scenarios (where interaction is needed) are rare; this is less pronounced on the sports dataset where the proportion of hard scenes (requiring interaction modelling) is larger, see Fig. 7. As for (ii), *in existing benchmark settings, people tend to walk toward their destinations mostly avoiding interactions*, except for the purpose of avoiding obstacles and collisions, whereas *interaction is an integral part of the activity in a sports scenario* such as playing a game of basketball.

Concurrently to our work, Chen et al. (2021)highlight the importance of the bias between training and testing environments and introduced a counterfactual analysis in which they intervene on the causal graph by cutting the edge between the predicted trajectory and the environment. To do so, they utilize several choices for the counterfactual trajectory such as the mean trajectory or a random trajectory. Our method, on the other hand, remove totally the influence of a feature (neighboring trajectory) by removing its edges in the interaction graph.

## E  INTERACTION GRAPH LEARNING

An interesting and related line of works is to explicitly learn an interaction graph as part of the trajectory prediction task. Kipf et al. (2018) introduce the NRI framework, a variational auto-encoder, where the learned latent code represents the underlying interaction graph of the system. Several recent works build upon the NRI framework by allowing the interaction graph to change over time (i.e., dynamic relations) (Li et al., 2020; Graber & Schwing, 2020). Löwe et al. (2020) leverage the shared knowledge across different samples and scenes and learn the causal relations between time series (e.g., moving agents). We adapted the latter work into the state-of-the-art framework for tra-

Table 3: Average errors (min-ADE / min-FDE) of different variants of the Traj++NRI framwork with different priors on the learned interaction graph (0.5, 0.9, 0.1).

| Method \Setup | With Interaction | Without Interaction |
|---|---|---|
| Traj++ | 0.30 / 0.54 | 0.30 / 0.54 |
| Traj++NRI (0.5) | 0.32 / 0.58 | 0.30 / 0.53 |
| Traj++NRI (0.9) | 0.31 / 0.55 | 0.30 / 0.55 |
| Traj++NRI (0.1) | 0.32 / 0.58 | 0.30 / 0.54 |

jectory prediction (Traj++ (Salzmann et al., 2020)) by replacing the edge encoder of the framework with a learned interaction graph (referred as Traj++NRI) and train the model on the ETH-UCY/univ scene. Tab. 3 shows an analysis of the Traj++ model with its original edge encoder and the learned interaction graph from Löwe et al. (2020) for different prior graphs. We observe that learning an interaction graph did not lead to a better model and dropping the edges during inference yielded similar/better performance. This supports our hypothesis that on common benchmarks, social interaction features are not learned.

## F  SCENE-AGENT INTERACTION

Although our work targets mainly the analysis between agents, extending this analysis to the interaction between dynamic agents and the scene (e.g., semantic map) is valuable. To this end, one option would be to learn a compact set of high-level map features (e.g., via a CNN encoder) and to add this as an additional single node to the interaction graph. Our approach could then be applied to quantify also the contribution of map information to model performance.

