# OpenReview forum: "You Mostly Walk Alone: Analyzing Feature Attribution in Trajectory Prediction"
_ICLR.cc/2022/Conference — ICLR 2022 Poster_

### Official Review · Reviewer_jvES · 2021-10-18

**Correctness:** 4
**Technical Novelty And Significance:** 3
**Empirical Novelty And Significance:** 3
**Recommendation:** 8
**Confidence:** 4

**Main Review:**

This paper proposes to analyze the contribution of different clues with the Shapley values and obtain an interesting conclusion that the past trajectory of the target is the only feature used for predicting its future for state-of-the-art trajectory prediction methods on standard benchmark datasets.

Strengths:
1) I think this conclusion makes sense. Some casual-inference based trajectory prediction methods [1] also found the interactions are biased in the current trajectory prediction systems and datasets. This paper proposes to use the Shapley values to analyze the contribution of different clues, which provides a quantitative evaluation of these biased interactions.
2) The contribution of this method is clear. The discovery of the uselessness of interactions can help other researchers to develop better trajectory prediction methods. The experimental results on SportVU are also an important discovery to show the trajectory prediction models have the capability to learn the interactions.  I guess it is because the interactions in the training and testing dataset on SportVU are not biased, while the interactions in ETH-UCY or SDD are based.
3）The paper is well written, which clearly shows the motivation, conclusion, and technical details. Compared with original Shapley values, this method modifies the value of the adjacency matrix as zero to drop features and proposes a local-global contribution aggregation strategy for variable features in different scenarios.

Weakness:
1) Some methods[1] also found the biased interactions and analyzed this in the perspective of causal inference (Effect of treatment on the treated). I think the differences between the two methods are clear, and there are the links of this method to causality in the discussion  Yet, it is suggested to provide the discussion with this paper, which also thinks the interactions are biased and analyzes this with the Effect of Treatment on the Treated.
2) In my opinion, some conclusions should be further discussed. For example, why SportVU is ok to learn interactions but not SDD?  I guess it is because of the level of interaction bias in training and testing environments.  SportVU only contains the scene of NBA for both training and testing. However, in other datasets, the scenes are changed, which is similar to the out-of-distribution problem. This setting may be more appropriate for self-driving. It is suggested to conduct a new experiment, where fewer interactions biases in training and testing environments. For example, all training and testing trajectories are from the ETH scene, but not from hotel, zara or univ.
3) Some technical details are suggested to provide. The performance of Traj++ on SDD shows a dramatic improvement than PECNet. Can you provide the details of the training and testing of Traj++ on SDD?

[1]Chen G, Li J, Lu J, et al. Human Trajectory Prediction via Counterfactual Analysis[C]//Proceedings of the IEEE/CVF International Conference on Computer Vision. 2021: 9824-9833.



**Summary Of The Paper:**

Different from attributing the trajectory prediction power with the success of modeling agent-agent interactions, this paper presents a new perspective that the past trajectory of the target is the only feature used for predicting its future for state-of-the-art trajectory prediction methods on standard benchmark datasets.  To analyze the contribution of different clues, this paper applies to compute the Shapley values. The proposed method modifies the value of the adjacency matrix as zero to drop features and analyze. Besides, a local-global contribution aggregation method is proposed for the problem of variable features in different scenarios.

This paper analyzes many state-of-the-art trajectory prediction models (Trajectorn++, PECNet, and Social-STGCNN) on many datasets (ETH-UCY, SDD, nuScenes, and SportVU). The experimental results on SportVU show the trajectory prediction models have the capability to learn the interactions on the datasets where interactions are more important.


**Summary Of The Review:**

Overall, I think it is a good paper with some minor problems that presents a new perspective that the past trajectory of the target is the only feature used for predicting its future for state-of-the-art trajectory prediction methods on standard benchmark datasets. On the balance of positive and negative points, I think it is a borderline paper and tend to accept.

Post Rebuttal： Thanks for the authors' feedback. The rebuttal answers most of my questions, which raises my score. It is suggested to further analyze why the interactions have limited effect in trajectory prediction. It is unintuitive but makes sense. There are some potential reasons in my views: 1) There are biases in interactions? 2) The multi-modality generation breaks the effect of interactions? 3) Is this finding generalized to rule-based methods, such as Social Force? I will further raise my score if these questions can be addressed.

---

> ### Author Response · Authors · 2021-11-19
> **Author Response to Reviewer jvES**
>
> ### Related work
> > ”It is suggested to provide the discussion with the paper ”Human Trajectory Prediction via Counterfactual Analysis””
>
> We thank the reviewer for pointing out a concurrent work published at ICCV’21 that also analyzes interaction features in the context of causal inference. We will add a discussion of this method to the revised manuscript.
>
> ### Training/testing bias
> > ”It is suggested to conduct a new experiment, where fewer interactions biases in training and testing environments”:
>
> We thank the reviewer for his valuable suggestion to gain more insights into the reasons for the behavior difference between existing benchmarks and the sports dataset. To achieve that, we split one of the scenes of the ETH-UCY into two subsets, train on one, and test on the other to reduce the interaction bias. We report 0.32/0.57 with interactions and 0.33/0.59 without interactions (min-ADE/min-FDE). We conclude that the interaction bias between train/test environments contributes only a little to our finding. We hypothesize two complementary explanations for this: (i) a large difference in the *frequency* of interactions, and (ii) a large difference in the *nature* of interactions. As for (i), the concurrent work “On Exposing the Challenging Long Tail in Future Prediction of Traffic Actors” highlights the long-tailed distribution of common pedestrian benchmarks (e.g., ETH-UCY) meaning that most scenarios can be predicted well using a simple linear Kalman filter (so-called easy scenarios) while more challenging scenarios (where interaction is needed) are rare; this is less pronounced on the sports dataset where the proportion of hard scenes (requiring interaction modeling) is larger, see https://ibb.co/GdWbpbt. As for (ii), in existing benchmark settings, people tend to walk toward their destinations mostly avoiding interactions, except for the purpose of avoiding obstacles and collisions, whereas *interaction is an integral part of the activity in a sport scenario* such as playing a game of basketball.
>
>
> ### Technical details
>
> The reported results on SDD for PECNet and Traj++ are not directly comparable as the errors are in different domains (in pixels for PECNet and in meters for Traj++). Thank you for the comment; we will clarify this in the revised version.

---

> ### Author Response · Authors · 2021-11-28
> **Thanks for the reply.**
>
> We thank the reviewer for the comments and glad to know that our rebuttal answered most of the raised questions. Below we discuss the additional questions about our finding:
>
> "There are biases in interactions": Yes, we agree with the reviewer. Existing traffic benchmarks have some rare cases of interactions with a large diversity, thus learning the underlying rules of interactions is much harder. This is also supported by the concurrent work "On Exposing the Challenging Long Tail in Future Prediction of Traffic Actors".
>
> "The multi-modality generation breaks the effect of interactions": According to our results, SOTA methods are able to learn interaction cues on a different domain (e.g., SportVU dataset) where the output is also multi-modal (i.e., multiple futures are generated). In this paper, we only claim that these methods cannot use interaction cues on the common traffic benchmarks where the interaction signal is too weak to be picked up.
>
> "Is this finding generalized to rule-based methods, such as Social Force": This is a good point. We conjecture that rule-based methods are not affected by our finding since the rules are designed to be generic and will generalize (to a certain degree). On the other hand, learning the interactions overfits on the hard traffic datasets. To validate this, we need to run one of the rule-based method (e.g., Social Force) and conduct the same analysis (via our Shapley values).
>
> We will add this discussion to the paper.

---

### Official Review · Reviewer_9atf · 2021-10-31

**Correctness:** 4
**Technical Novelty And Significance:** 4
**Empirical Novelty And Significance:** 4
**Recommendation:** 10
**Confidence:** 5

**Main Review:**

I believe this work is valuable overall for the trajectory prediction community. They present the problem well and the general framework of trajectory prediction[history encoder, interaction encoder and decoder] is workable and applied to most of the current works.
Question: The choice of using static, non-interacting agents seems feasible but what about other options? Can the authors elaborate on this and add a discussion for other choices and how these choices could alternate the results? For example, one might suggest using the slowest agent.

It seems from the Shapley values that the random agents doesn't contribute much to the results because of the lack of interaction as shown in the work and this is a good finding.

The only thing I’m skeptical about is the definition of the social interaction score. Though it seems justified by the presented argument, it needs a kind of empirical verification. Training from scratch the trajectory prediction models with no interactions (aka single agents only, no neighbours) should produce similar (or better) results to the ones with the neighbours. This will prove the feasibility of the social score and the whole findings of the papers. I can understand that the training can take a while, but providing results on two models with at least a subset of the ETH/UCY (I’d avoid the ETH and use Zara1, Zara2, Hotel as ETH is highly unbalanced) dataset will be a strong addition.

=== Post Rebuttal Comment ===
I believe that the authors have answered the concerns raised by the reviewers and my concerns. The new experiments on Trajectron++ proved the theoretical analysis. Overall, in the past years since 2016 plenty of deep models were introduced to solve the trajectory prediction problem with emphasis on the "Social Interaction" between the agents. This paper proves via analysis and empirical results that such "Social Interaction" does not exist or have a minimal effect. Thus, I find this work significant in this area as it will lead future research into figuring out a true form of pedestrian’s interaction. Based on this, I’m raising my score.


**Summary Of The Paper:**

This work discusses the features that attribute to the prediction results in trajectory prediction models. They apply a Shapley values to attribute the features and results, attempting to answer a question if the social models are really social or not. The meaning of a model being social in this case is that the model can use neighbour features in order to predict the trajectory. The work defines a measurement metric called the social interaction score, in which it measures how well the interaction between agents attributes to the prediction of the other agent trajectory.


**Summary Of The Review:**

The paper is valuable to the community of trajectory prediction models. It directly discusses if the “social” interaction exists or not in previously reported models. Some questions were raised regarding the choice of the static, non-interacting agent and the feasibility of the presented social interaction score.

---

> ### Author Response · Authors · 2021-11-19
> **Author Response to Reviewer 9atf**
>
> ### Choice of baseline
> >”The choice of using static, non-interacting agents seems feasible but what about other options?”
>
> Choosing the baseline to be a static non-interacting agent means that when dropping a feature (e.g., interaction feature with a neighboring agent), we remove that agent by cutting the edge in the interaction graph. Other alternatives are also possible (e.g., replacing a neighboring agent by a random or slower one); however, in all these cases the baseline agent would still contribute some signal that could affect the obtained scores. This is why we opted for cutting edges in the graph in our variant which ensures that the baseline agent cannot possibly influence the predicted trajectory for the target agent.
>
>
> ### Social interaction score
> > ”Training from scratch the trajectory prediction models with no interactions (aka single agents only, no neighbours) should produce similar (or better) results to the ones with the neighbours”
>
> We thank the reviewer for his valuable suggestion which is quite similar to the point raised by reviewer `hYNN`. To validate this, we conducted an additional experiment where we train on one of the scenes of ETH-UCY (univ) without interaction and compare its error to the one trained with interaction. We observe that both models yield the same error (0.30/0.54 for min-ADE, min-FDE), which is consistent with and further supports our finding that interactions are hardly used to make predictions on this dataset.

---

> > ### Comment · Reviewer_9atf · 2021-11-19
> > **Thanks for the response**
> >
> > I believe if you are able by now to obtain the Univ datset results on Trajectron++ you can have it on the full set(ETH/UCY) to confirm the findings? Can this be reported during this period?

---

> > > ### Author Response · Authors · 2021-11-22
> > > **Thanks for the reply.**
> > >
> > > We thank the reviewer for the comment. We also report the results on all scenes of the ETH-UCY dataset, please see Table 2 in the revised paper under appendix C. Overall, we observe that training the Traj++ model with/without interactions yields the same performance.

---

### Official Review · Reviewer_dgLJ · 2021-11-01

**Correctness:** 4
**Technical Novelty And Significance:** 3
**Empirical Novelty And Significance:** 3
**Recommendation:** 6
**Confidence:** 4

**Main Review:**

Strengths:
1. The paper is generally well written and easy to follow. The authors provided sufficient background information and a relatively comprehensive literature review. The major motivation of this work is clearly stated.
2. Evaluation and analysis of the interaction modeling of trajectory prediction models is an interesting task and could contribute to figuring out the most useful approaches/representations to capture mutual interactions.
2. In order to support the authors' claims, they conducted many experiments and the results seem supportive based on the assumption that the quantifiable metric they proposed is reasonable.

Weaknesses:
1. Since all the experimental results are based on the Shapley value, it would be better to provide more explanation and evidence for its applicability in the area of trajectory prediction. Also, is there any limitation/caveat when using Shapley value as the metric for this purpose?
2. Three closely related papers are listed below, which explicitly model the interactions/relations between interactive agents. It would be better to include them in the literature review. In particular, it would be better to conduct some experiments using the model proposed in [1], which makes the statement even more convincing.
[1] Neural relational inference for interacting systems, ICML 2018.
[2] EvolveGraph: Multi-Agent Trajectory Prediction with Dynamic Relational Reasoning, NeurIPS 2020.
[3] Dynamic Neural Relational Inference, CVPR 2020.


**Summary Of The Paper:**

This paper focuses on analysis on whether the existing trajectory prediction methods indeed capture the interactions between multiple agents and uses two example methods on multiple benchmark datasets to obtain supportive results for their claims. More specifically, the contributions are three folds:
1. This paper addresses feature attribution for trajectory prediction to gain insights about the actual cues contemporary methods use to make predictions. The authors designed a variant of Shapley values that is applicable to a large set of trajectory prediction models.
2. This paper quantifies feature attributions both locally and globally and studies the robustness of given models.
3. This paper claims that existing models do not use interaction features, which is contrary to the statements made by the authors of the original papers.

**Summary Of The Review:**

Overall, I think that the contributions of this paper are interesting and could raise more discussion and exploration on the interaction modeling in multi-agent systems. I have a positive feeling about this paper. My major concern is the suitability of the proposed Shapley value metric in the task of trajectory prediction. Hope to see more explanations in the rebuttal.

---

> ### Author Response · Authors · 2021-11-19
> **Author Response to Reviewer dgLJ**
>
> ### Shapley values limitation and applicability
> > ”It would be better to provide more explanation and evidence for its applicability in the area of trajectory prediction”
>
> As explained in Section 2.3, several choices are involved in designing a feature attribution method that makes sense for a particular problem at hand. Our general formulation of the problem (Figure 2), particularly introducing the concept of interaction graph, allows us to drop a feature (e.g., neighboring agent) by cutting its edge, thus facilitating the adaptation of Shapley values to our task.
>
> > “Is there any limitation/caveat when using Shapley value as the metric for this purpose?”
>
> Different from current variants like baseline/marginal/conditional Shapley values, our variant ensures that there is no signal that can affect the obtained scores. As discussed at length in section 5, one caveat is that care needs to be taken when trying to attach a causal interpretation to feature attribution scores (whether obtained via Shapley values or through other attribution methods).
>
> ### Missing related works
> > ”It would be better to conduct some experiments using the model proposed in [1]”
>
> Thank you for the pointer; we will add a discussion thereof to the revised paper. Explicitly learning an interaction graph as reported in the NRI paper is an interesting direction. We adapted a recent work that builds on top of the NRI paper, ”Amortized Causal Discovery: Learning to Infer Causal Graphs from Time-Series Data”, which aims at learning the causal relations between different time series (e.g., moving agents in our task) by leveraging shared knowledge across different samples and scenes. In particular, we replaced the edge encoder of the Traj++ model with the learned interaction graph of the ”Amortized” paper and train on the ETH-UCY/univ scene, see https://ibb.co/JphfbXq for the obtained results. Note that we experiment with different priors on the interaction graph (0.5, 0.1, 0.9). These results show that learning an interaction graph does not lead to a better model and dropping the edges during inference yields similar/better performance. This supports our hypothesis that on common benchmarks, social interaction features are not learned. We will add a small section to the supplementary material discussing these experiments.

---

> > ### Comment · Reviewer_dgLJ · 2021-11-29
> > **Thanks for the response!**
> >
> > I would like to thank the authors for their response and update in the new version. Most of my concerns and comments have been addressed.

---

> > > ### Author Response · Authors · 2021-11-29
> > > **Thanks for the feedback**
> > >
> > > We thank the reviewer for the feedback and glad to know that our rebuttal addressed most of the raised concerns.
> > >
> > > We kindly ask the reviewer to provide a final post-rebuttal comment about our paper and its recommendation.

---

### Official Review · Reviewer_ef1N · 2021-11-02

**Correctness:** 4
**Technical Novelty And Significance:** 2
**Empirical Novelty And Significance:** 4
**Recommendation:** 8
**Confidence:** 4

**Main Review:**

This paper is presented clearly and a better feature attribution method can help us understand blackbox predictors and make sure if the presented model learns the interactions we want. I only have a few questions.

Novelty. Changing latent codes or features and computing their influence on performance is not entirely new. It has been used to understand blackbox vision/language models. The authors need to compare with prior work and show why this proposed measure is novel rather than applying it to trajectory prediction.

Extending to predictors that include map info. The way to drop features and aggregate the Shapley value is defined purely based on the agent interaction graph. How does this extend to other approaches that use information such as map features?

Choice of behavior for computing the Shapley values. The contribution of the other agents does not necessarily influence the accuracy of the predicted accuracy, it may reduce the uncertainty of behaviors too. How does this paper handle the influence not directly reflected in accuracy?

-----
Post-rebuttal update:
The authors answered my questions and address other reviewers' questions. While the proposed approach directly extends Shapley value for trajectory prediction, the analysis is valuable for the community. So, I’m raising my score.

**Summary Of The Paper:**

This paper proposes assessing the contribution of each feature by computing a version of Shapley values based on the existing trajectory predictor formulation. They apply this measure to the state-of-the-art trajectory prediction methods and found that many standard benchmarks don’t require reasonings of agent interactions.

**Summary Of The Review:**

The paper clearly presents a variant of Shapley value for trajectory prediction tasks. The proposed measure helps identify if a model uses a feature for prediction. My main question is about the novelty of the proposed approach.

---

> ### Author Response · Authors · 2021-11-19
> **Author Response to Reviewer ef1N**
>
> ### Novelty
> > ”The authors need to compare with prior work and show why this proposed measure is novel rather than applying it to trajectory prediction”
>
> Shapley values are a well-known and established technique for feature attribution, and we agree that they have previously been used in different domains. As explained in Section 2.3, however, several choices are involved in designing a feature attribution method that makes sense for a particular problem at hand, and we believe we are the first to propose a variant specifically for the context of trajectory prediction (or, more generally, for attribution in the presence of interaction graphs). Although comparing different feature attribution methods is an interesting topic, it is not the main aim of the paper; our choice of Shapley values (over other attribution techniques) is justified by being the only method satisfying a desirable set of axioms (see, e.g., “The many Shapley values for model explanation” ICML. 2020).
>
> Regarding novelty, we also kindly point the reviewer to our extensive experimental evaluation and (perhaps surprising) empirical findings in analyzing several recently published trajectory models, as well as to the newly proposed social interaction score and robustness analysis.
>
>
> ### Extension to other features
> > ”How does this extend to other approaches that use information such as map features?”
>
> This is an interesting point and clearly a possible direction for further research. While the main scope of the paper is to analyze interactions between agents (since this was claimed to be one of the key drivers of recent improvements in the field of trajectory prediction), we agree that analyzing the interaction between an agent and the scene is also valuable. To this end, one option would be to learn a compact set of high-level map features (e.g., via a CNN encoder) and to add this as an additional single node to the interaction graph. Our approach could then be applied to quantify also the contribution of map information to model performance. We thank you for the suggestion and will add the above discussion to the revised manuscript.
>
>
> ### Choice of the target function
> > ”How does this paper handle the influence not directly reflected in accuracy?”
>
> Using uncertainty as a target function is indeed another option if the objective is to understand the influence of neighboring agents on predictive uncertainty. In principle, our approach (particularly, the choice of baseline) is agnostic to the target function. We focus on model performance as many SOTA models (e.g., PECNet) do not offer probabilistic outputs for which uncertainty can easily be measured. On the other hand, using NLL as the target function implicitly also takes the uncertainty of the prediction into account since the NLL is smaller when the predicted distribution is closely centered about the true trajectory (i.e., the associated uncertainty is small).

---

> > ### Comment · Reviewer_ef1N · 2021-11-28
> > **Thanks for the response!**
> >
> > I thank the authors for the response and the updates to the paper! That answers my questions.

---

### Official Review · Reviewer_hYNN · 2021-11-03

**Correctness:** 3
**Technical Novelty And Significance:** 3
**Empirical Novelty And Significance:** 3
**Recommendation:** 5
**Confidence:** 4

**Main Review:**

The manuscript has good clarity and explains its contributions well.
The analysis of the evaluation aligns with the claim that models are not learning interactions from ETH/UCY, nuScenes, and SDD, but are learning interactions from SportVU.
The evaluation of HTP models is an interesting research area that has not received nearly as much attention as works producing new HTP models using the same traditional evaluation metrics as others (i.e., minADE and minFDE).

When dropping a subset of agents to determine the influence on model performance, the subset is replaced with static agents that are assumed not to influence the remaining agents. However in practice, static, non-interacting obstacles are still considered by humans during navigation due to collision avoidance.
To my understanding, the modified version of baseline Shapley values does not completely ensure that there is no interaction.

In Section 3.3, the authors note that random agents should not contribute to the future of the agent-of-interest, but again, they still play a role in the movement of the agent-of-interest due to collision avoidance.
Yet, in the results reported in Figure 4, random agents contribute at times negatives Shapley values (e.g., Traj++Edge on ETH/UCY, nuScenes, and SDD).
Also, what is the interpretation of the negative contributions of the random agent and interaction (specifically for Traj++Edge on ETH/UCY)?

In Figure 4, the Shapley values reported differ for each dataset. ETH/UCY is missing PECNet, nuScenes is missing STGCNN and PECNet, SDD is missing STGCNN, and SportVU is missing STGCNN.
Also, the Shapley values for ETH/UCY, nuScenes, and SDD are reported based on NLL, but not on min-ADE with the exception of subfigures (d) and (f).

Regarding the interactions in ETH/UCY and nuScenes, there are numerous scenes in which no interaction occurs between agents (especially in nuScenes).
It would be helpful to see the contribution of social interactions on the subset of scenes in each dataset that have meaningful social interactions present in them.

It is surprising to see that the same models show significantly higher contributions of interaction in SportVU than in ETH/UCY, which has hundreds of scenes under "students001" and "students003" and feature extremely high levels of interaction between many pedestrians.

The authors note that the contributions of past trajectory and neighbors can be compared for an individual model (e.g., for Traj++Edge), but do not explain how the contributions can be compared between models (e.g., between PECNet and Traj++).
According to Figure 4(f), the contribution of past trajectory to PECNet is 3 to 4 times larger than the contributions of past trajectory to Traj++ and Traj++Edge.
What does this mean?

The authors note that the interaction encoder of Traj++Edge is stronger, so it has higher contributions of history and interaction on SportVU, but for ETH/UCY, the contribution of interaction is less than Traj++ and is negative.

The PECNet paper reports results on ETH/UCY, but the results have not been included into Table 1.
The results of PECNet and Traj++ differ from the results reported in their respective papers. The authors should make a note of why they differ and also clarify on the missing data.

I have some concern about the results reported in Table 1 for models without interaction. The ideal way to report this result is to train on a modified dataset in which every scene with $n$ agents is split up into $n$ subscenes with one agent each, thus ensuring that there is no possibility of agent interaction.

It would be helpful to see the local analysis on SportVU also done for interactive and non-interactive scenes in ETH/UCY, nuScenes, and SDD to showcase the lack of interaction learning.


**Summary Of The Paper:**

This manuscript addresses the difficulty in interpreting what information a model learns from human trajectory prediction datasets. The authors propose a feature attribution method for trajectory prediction based on Shapley values. The authors also show that for commonly used datasets such as ETH/UCY, SDD, and nuScenes, models Trajectron++ and PECNet do not learn interaction information. However, on datasets featuring more interaction, such as SportVU, these models are able to learn social interactions.


**Summary Of The Review:**

Although I believe that this manuscript is well-written, tackles an interesting problem, and has an evaluation that is consistent with its claims, I do not think there is enough evidence to back the strong claim that SOTA HTP models are unable to learn almost any interactions for well-studied datasets that have cases of high interaction, but are able to learn interactions for a dataset with more interaction.

---

> ### Author Response · Authors · 2021-11-19
> **Author Response to Reviewer hYNN (part 1 of 2)**
>
> ### Dropping interaction features
> >”When dropping a subset of agents to determine the influence on model performance, the subset is replaced with static agents that are assumed not to influence the remaining agents”
>
> There appears to be a misunderstanding and we are sorry for not clearly communicating that we do not replace dropped agents with static agents but instead remove them entirely from the scene. To achieve that, we cut their outgoing edges in the interaction graph (figure 2C), thus eliminating any influence they may have on the predicted future of the target agent (see page 5, line 4).
>
>
> ### Random agents
> >”Random agents still play a role in the movement of the agent-of-interest due to collision avoidance”
>
> This is a valid argument in the context of pedestrian navigation. However, in practice, these are rather less frequent cases, and generally, the random agent should not contribute to the target agent (we can also call it a ghost agent). To tackle this concern, one idea is to restrict the random agent to be not in close proximity to the target agent. Thank you for pointing out this case; we will discuss it in the paper (section 3.3).
>
> ### Negative contribution
> >”What is the interpretation of the negative contributions”
>
>  If the contribution of a neighboring agent is negative, this means that the prediction error for the target trajectory increases by taking that agent into account. An example is shown in Figure 5b, where agent 6 moves to the right while the actual future of the target agent is to change direction and go left (leading to a negative contribution for agent 6). We will extend our local analysis section to clarify the interpretation of negative contributions.
>
>
> ### Missing models/metrics:
> In our evaluation, we rely mainly on the provided source code of the corresponding works where every model has its own pre-processing procedure of the used datasets. For instance, PECNet is not evaluated on nuScenes, and STGCNN is not evaluated on SDD. Bringing these datasets into the format acceptable for these models is non-trivial. As for the min-ADE on the Traj++ and social-STGCNN, we report these results in Table 1, supporting the same conclusion as the NLL figures (4a). We will add the corresponding figures to the revised paper for completeness.
>
>
> ### Evaluation on a subset of scenes
> >”It would be helpful to see the contribution of social interactions on the subset of scenes in each dataset that have meaningful social interactions”
>
> This is a good point, but we need additional annotation to filter scenes with meaningful social interaction (e.g., which sample has a significant social interaction). However, following the reviewer, we use our social interaction score and report a breakdown obtained on the ETH-UCY for the best model (Traj++Edge), see https://ibb.co/WfmxPVd (note that the y-axis uses logarithmic scaling). 1.4% of all samples have social interaction scores larger than 0.5 based on the NLL metric.
>
>
> ### Surprising finding
> >”It is surprising to see that the same models show significantly higher contributions of interaction in SportVU than in ETH/UCY”
>
> We hypothesize two complementary explanations for this: (i) a large difference in the *frequency* of interactions and (ii) a large difference in the *nature* of interactions. As for (i), the concurrent work “On Exposing the Challenging Long Tail in Future Prediction of Traffic Actors” highlights the long-tailed distribution of common pedestrian benchmarks (e.g., ETH-UCY) meaning that most scenarios can be predicted well using a simple linear Kalman filter (so-called easy scenarios) while more challenging scenarios (where interaction is needed) are rare; this is less pronounced on the sports dataset where the proportion of hard scenes (requiring interaction modelling) is larger, see https://ibb.co/GdWbpbt. As for (ii), in existing benchmark settings, people tend to walk toward their destinations mostly avoiding interactions, except for the purpose of avoiding obstacles and collisions, whereas *interaction is an integral part of the activity in a sports scenario* such as playing a game of basketball.
>
>
> ### Models comparison
> >”How the contributions can be compared between models”
>
>  A direct quantitative comparison between models is not straightforward as the estimated Shapley values refer to performance change in different models and are thus not necessarily on the same scale. However, looking at Figure 4f, we can roughly conclude that PECNet relies more on the past trajectory than Traj++. Another interesting comparison is shown in Figure 4e, where Traj++Edge relies more on the past and interaction features than Traj++ which is explained by the design of our Traj++Edge (section 4.1) where both the history and interaction encoders are made stronger.

---

> ### Author Response · Authors · 2021-11-19
> **Author Response to Reviewer hYNN (part 2 of 2)**
>
> ### Missing data/different results
>
> Although PECNet reports its results on ETH-UCY, they don’t provide the training setup (e.g., hyperparameters) used for ETH-UCY with social pooling or pre-trained models. Since social pooling is the only component used for interaction modeling in PECNet, it isn’t very meaningful to report its results in Table 1. Moreover, our numbers for PECNet on SDD are slightly different (and even better for min-ADE 9.96 vs 9.29) from the original paper, obtained by retraining their model using their source code. We will clarify these details in the revised paper. Regarding results on Traj++, our numbers are obtained after fixing an issue with the derivative computation reported to the corresponding authors who promised to revise their paper. We will include a note about that in the paper.
>
>
> ### Retraining on dataset without interaction
> > “The ideal way to report this result is to train on a modified dataset in which every scene with n agents is split up into n subscenes with one agent each, thus ensuring that there is no possibility of agent interaction”
>
> This is a good point. Although our goal is to evaluate how existing models trained on benchmarks failed to learn interaction, the proposed alternative setup is also meaningful. Therefore, we retrain Traj++ on the Univ dataset without interaction (by considering agents without neighbors) and compare the results to the original model trained with neighbors. The errors of the two models are the same 0.30/0.54 (min-ADE/min-FDE). This further supports our main findings that these models trained on existing benchmarks (e.g., ETH-UCY) do not utilize interaction features.

---

### Author Response · Authors · 2021-11-19
**General Response to all Reviewers**

We thank the reviewers for their time and insightful comments.

The reviewers have questions mainly regarding some of the assumptions underlying our method (in particular, the choice of the target function and baseline), the applicability of Shapley values for trajectory prediction, and additional explanations of our findings. We address all of these in detailed comments to each reviewer.

To this end, we conducted the following **additional experiments**:
- To further support our findings, reviewers `hYNN` and `9atf` suggested retraining one of the SOTA models from scratch on a dataset without interaction by removing neighbors to assess if models need interaction or not. We selected the Traj++ model, retrained it on the ETH-UCY/univ dataset, and compared the results to the same model trained with interactions. We observe that both models (with and without interaction) yield the same error (0.30/0.54 for min-ADE/min-FDE), which is consistent with our findings. We will add this analysis on the remaining scenes of the ETH-UCY dataset to the revised paper.

- To gain more insight into the finding that models behave differently between existing benchmarks and the SportVU dataset, reviewer `jvES` suggested splitting one of the scenes of ETH-UCY into two subsets, training on one, and testing on the other to reduce interaction bias. We report 0.32/0.57 with interactions and 0.33/0.59 without interactions (min-ADE/min- FDE) and conclude that interaction bias between the train and test environments only makes a negligible contribution to our findings. We hypothesize two complementary ex- planations for this: (i) a large difference in the frequency of interactions, and (ii) a large difference in the nature of interactions. As for (i), the concurrent work “On Exposing the Challenging Long Tail in Future Prediction of Traffic Actors” highlights the long-tailed distribution of common pedestrian benchmarks (e.g., ETH-UCY) meaning that most scenarios can be predicted well using a simple linear Kalman filter (so-called easy scenarios) while more challenging scenarios (where interaction is needed) are rare; this is less pronounced on the sports dataset where the proportion of hard scenes (requiring interaction modelling) is larger, see https://ibb.co/GdWbpbt. As for (ii), in existing benchmark settings, people tend to walk toward their destinations mostly avoiding interactions, except for the purpose of avoiding obstacles and collisions, whereas interaction is an integral part of the activity in a sports scenario such as playing a game of basketball.

- To further validate our findings, reviewer `dgLJ` suggested exploring the interesting direction of explicitly learning an interaction graph as reported in the NRI paper. We adapted a recent work that builds upon the NRI paper, ”Amortized Causal Discovery: Learning to Infer Causal Graphs from Time-Series Data”, which aims at learning the causal relations between different time series (e.g., moving agents in our task) by leveraging shared knowledge across different samples and scenes. In particular, we replaced the edge encoder of the Traj++ model with the learned interaction graph of the ”Amortized” paper and train on the ETH-UCY/univ scene, see https://ibb.co/JphfbXq for the obtained results. Note that we experiment with different priors on the interaction graph (0.5, 0.1, 0.9). These results show that learning an interaction graph did not lead to a better model and dropping the edges during inference yielded similar/better performance. This supports our hypothesis that on common benchmarks, social interaction features are not learned. We will add a section to the supplementary material discussing these experiments.

We are happy to answer any additional questions the reviewers might have.

---

### Author Response · Authors · 2021-11-22
**General Response to all Reviewers**

We thank again the reviewers for their time and feedback.

Please be aware that we updated the manuscript by including new experiments and addressing some of your comments. The difference is highlighted in yellow. The main changes are given in appendices C, D, E, and F.

---

### Decision · Program_Chairs · 2022-01-20

**Decision:**

Accept (Poster)

**Comment:**

The manuscript brings up an important issue: that current methods and datasets don't generally highlight interactions when it comes to trajectory prediction. This is despite the fact that it would seem that current methods incorporate agent interactions and that datasets appear to require reasoning about agent interactions. This qualitative and quantitative observation should lead to better datasets in the future as well as more refined metrics pushing the field forward. Reviewers were in agreement that this is a strong submission. The authors responded with substantive new experiments that cleared up any lingering issues.